# Neural Correlates of Language Models Are Specific to Human Language

**Iñigo Parra**
Department of Linguistics
University of California, Berkeley
Berkeley, CA 94702
iparra@berkeley.edu

## Abstract

Previous work has shown correlations between the hidden states of large language models and fMRI brain responses, on language tasks. These correlations have been taken as evidence of the representational similarity of these models and brain states. This study tests whether these previous results are robust to several possible concerns. Specifically this study shows: (i) that the previous results are still found after dimensionality reduction, and thus are not attributable to the curse of dimensionality; (ii) that previous results are confirmed when using new measures of similarity; (iii) that correlations between brain representations and those from models are specific to models trained on human language; and (iv) that the results are dependent on the presence of positional encoding in the models. These results confirm and strengthen the results of previous research and contribute to the debate on the biological plausibility and interpretability of state-of-the-art large language models.

## 1 Introduction

Transformer language models (LMs) have reached near-human accuracy on a broad spectrum of linguistic benchmarks, largely by scaling parameters, data, and compute [26]. Whether the internal computations that support this success reflect the neural dynamics of human language processing, however, remains unresolved. Prior work has demonstrated voxel-wise correlations ("brain scores") between LM hidden states and fMRI responses during reading comprehension [41, 20]. Yet some fundamental questions persist. First, it is unclear wether these correlations are specific to human language or if they arise from generic statistical artifacts rather than from exposure to natural language. Second, the contribution of different architectural components of transformer models is still debated. These can offer new alternatives to test the significance and accuracy of the brain-model linguistic correlations. For example, if the correlations are natural language specific, a significant drop in performance would be expected from the ablation of positional (structural) information.

We address these issues through a comprehensive transformer-brain comparison using fMRI data [37] and 19 *text*-trained transformers. In addition, we contrast these correlations with protein-trained controls that share architecture and objective but lack linguistic input. Beyond standard correlation-based scores, we compute unbiased centered-kernel alignment (CKA) and Gromov-Wasserstein (GW) distances, providing complementary geometric views of representational similarity.

The results reveal three key findings. Removing positional encodings drops brain scores by up to 0.4 ($r$), inflates GW distance, and flattens CKA curves. In addition, PCA to 50 principal components leaves the core effects intact, indicating that alignment is not driven by sheer dimensionality. Finally, replacing LMs with models trained on non-linguistic sequences vanishes alignment, ruling out task-generic explanations.

Preprint.

## 2 Related Work

[57] carried out the first quantitative ANN-to-single-neuron comparison by training a multilayer back-propagation network on a sensorimotor transformation and demonstrating that its hidden units' tuning curves matched macaque posterior-parietal neurons. Subsequent work extended this agenda into unsupervised learning, showing that a mutual-prediction network developed disparity-selective units akin to those in early visual cortex [6]. [39] introduced a biologically inspired hierarchy of simple and complex units. This work demonstrated that the model's intermediate and output stages aligned with ventral-stream responses in V4 and inferotemporal cortex.

The introduction of deep learning in the early 2010s, and language models (LMs) more recently, has broadened the focus from vision to language. Recent work has used representational alignment methods to map deep language models onto neural data. [1] introduced Representational Stability Analysis (ReStA) to probe layer-wise robustness in transformer LMs and compared those representations to fMRI activations during story reading. [11] and [19] applied RSA and decoding frameworks to fMRI and MEG measurements, revealing where and when sentence-level representations emerge in the brain. Integrative predictive-processing models have shown that next-word prediction objectives yield representations that align closely with ECoG and fMRI data in language-selective regions [41, 20]. Studies of semantic topographies have reconstructed cortical maps of meaning from continuous speech, demonstrating that internal and hybrid semantic network models capture hierarchical semantic structure in the cortex [24, 55]. Similar work has shown that individual attention heads in transformer architectures reflect functionally specialized cortical gradients [30].

Parallel efforts have refined encoding and decoding frameworks for naturalistic stimuli. Previous studies have used deep-model activations to predict sentence-comprehension fMRI responses, highlighting variability in processing hierarchies across studies [3]. Similar work has dissected which task-specific representations best predict regional fMRI activity during reading versus listening [35]. [36] demonstrated that untrained models already capture baseline neural similarity but that training further improves brain scores across architectures [36]. Studies on training regimes and scaling show that instruction-tuning boosts brain alignment without extending behavioral gains [4], that alignment grows with model size but saturates around human-level linguistic competence [2], and that brain-predictivity follows scaling laws with parameter count and exhibits left-hemisphere dominance [8].

## 3 Methodology

### 3.1 fMRI Dataset

We used the dataset provided by [37] to compare brain and language model representations. This is consistent with previous studies [41, 19, 42, 35, 4][1]. For the brain model comparisons, we used the fMRI data from 6 adult subjects. The stimuli sentences consisted of simple and natural statements ($N_{stimuli} = 243$), with an average length of 13 words ($\sigma = 2.9$). The subjects were indicated to read the sentences naturally and thinking about their meaning. The brain activity was recorded while subjects were presented the sentences one at a time. For each subject, the dataset provides 243 BOLD signals represented as $\approx$200,000-dimensional vectors.

During preprocessing, we first selected the language regions-of-interest (ROIs) using the AAL parcellation atlas [49]. This was motivated by the potential limitations of the parcel selection procedure in previous work, which used an "independent localizer task" [41, p. 3]. Recent work has suggested fMRI's poor temporal resolution and the use of static, group-constrained ROIs may obscure rapid, syntactic-semantic operations [34]. Through the proposed methodological alternative, we aim to ensure consistency across participants on a network known to be heterogeneous and dynamic across individuals.

We studied the inferior frontal gyrus (IFG), superior temporal gyrus (STG), temporal pole (TP), and middle frontal gyrus (MFG). Additionally, we included the right hemisphere counterparts of the IFG, STG, and angular gyrus (AG). The selected ROIs have been shown to be recruited during syntactic processing and semantic retrieval [left IFG, left TP; 18], verbal working memory [left MFG; 53, 22],

---

[1]Dataset available at `https://osf.io/crwz7/`.
 Code: `https://github.com/IParraMartin/NCLM-Code`

lexical-semantic retrieval [left MFG; 53, 22], pragmatic processing [right AG; 44, 38], intelligible speech perception [left STG; 43, 23], and pitch and prosodic aspects of speech [right STG, right IFG; 43, 23].

For voxel selection, we performed split-half reliability selection [46] and considered the top 10% most reliable voxels for analysis. This allowed to select the most consistent and active voxels of the raw BOLD signals across participants.

## 3.2 Models and Activations

To align with the unimodal design of the fMRI task, we analyzed text-only transformer language models of two architectural types: *bidirectional* and *causal*[2].

Bidirectional models are optimized for masked-language modeling (MLM): roughly 25% of the input tokens are replaced with a `[MASK]` token and the model learns to predict the masked items from full left- and right-context. Because the network observes both past and future tokens, this training regime is less faithful to human language processing. During inference on the fMRI text stimuli, we recorded layer-wise activations of the special `[CLS]` token and retained attention matrices and head-wise outputs for further analysis.

Causal models are trained for next-word prediction (NWP). A causal mask in the self-attention mechanism restricts each token to attend only to earlier positions in the sequence, enforcing left-to-right processing [50]. Since causal models lack a `[CLS]` token, we derived sentence-level representations by mean-pooling the token embeddings at each layer. As with the bidirectional models, we stored the corresponding attention maps and head-specific activations.

For both model classes we repeated the extraction procedure after **zeroing the positional encodings**. Removing positional information prevented the network from exploiting explicit sequence structure, thereby attenuating syntactic cues in the representations. After the activation extraction procedure, the dataset included sentence level activations, head-wise activations, and attention matrices for both positional ([+POS]) and non-positional ([-POS]) conditions.

## 3.3 Correlation Analysis

The methodology used for the correlation analysis approximated that used by previous authors [25, 19, 48, 41, 10]. We included several modifications that aim to make the analysis more robust.

For every model, participant, and layer (or attention head), we standardized the model activations and fMRI responses within the training data of each cross-validation fold. Using 5-fold (i.e., five 80/20 splits) cross-validation, we fit a Ridge regression ($\lambda = 0.2$; best across folds obtained via grid search) on the training fold and predicted the held-out fold. We computed voxel-wise Pearson correlations ($r$) between predicted and observed activity, averaged them across voxels to obtain a fold-level correlation, and combined voxel $p$-values within each fold via Fisher's method. The final metric was the mean correlation and the median of the fold-level $p$-values. The brain score was obtained by normalizing the correlation metric by the estimated noise ceiling $(0.32)$[3]. This procedure was repeated for every participant and model, as well as positional condition to obtain layer-wise (and head-wise) scores.

To analyze the potential impact of the curse of dimensionality (CoD) on the correlation-based brain scores, we recomputed the brain scores for [+POS] and [-POS] conditions with PCA-reduced fMRI data. We used the first 50 principal components of the original signal, which preserved $\approx$75-80% of the original variance (reduction factor of $\times 4000$). With the results from both, PCA and non-PCA data, we assessed the impact of dimensionality in the brain scores.

## 3.4 Topological Analysis of Brain and LM Layers

To address previous criticism on the over-reliance on correlation-only results [e.g., 17], we compared the topological and geometrical properties of the brain signals and model layers. To do this, we compared the structural correspondence of the most "brain-aligned" model's internal representations

---

[2]Complete model specifications available in Appendix A.

[3]The ceiling was validated through a leave-one-out noise estimation.

(`opt-2.7b`) to all participants' brain signals. We computed the similarity of both (silicon and biological) data structures through the Gromov-Wasserstein (GW) distance. GW has been previously used in the context of object matching [32], subject-wise brain-signal alignment [47, 45], multi-modal clustering [21], color-perception comparisons between humans and LMs [27], and cell-development studies [28]. Recent results have also shown that GW provides structural distances that are not captured by simple correlations [5].

At its core, GW distance generalizes the optimal transport (OT) by seeking a soft-matching $T \in \Pi(\mu, \nu)$ between two sets of points $X$ and $Y$ that minimizes the discrepancy between the within-space pairwise distances (Equation 1). Here, $\mu, \nu$ are measures on $X$ and $Y$ with nonnegative weight vectors $p \in \mathbb{R}^n$ and $q \in \mathbb{R}^m$, $C_1 \in \mathbb{R}^{n \times n}$ and $C_2 \in \mathbb{R}^{m \times m}$ are the corresponding representational dissimilarity matrices (RMDs). Each entry $C_{1_{ik}}$ is the distance between point $i$ and point $k$ in $X$, and $C_{2_{jl}}$ each entry of points $j, l$ in $Y$. Using the quadratic loss $\mathcal{L}(a, b) = |a - b|^2$, GW is defined by:

$$
GW(\overbrace{C_1, C_2,}^{\text{RDMs}} \underbrace{p, q}_{\text{Weights}}) = \min_{T \in \Pi(p,q)} \sum_{i,k} \sum_{j,l} \overbrace{|C_{1,ik} - C_{2,jl}|^2}^{\text{Cost}} \underbrace{T_{ij} \, T_{kl}}_{\text{Joint}} \tag{1}
$$

where the feasible set

$$
\Pi(p, q) = \underbrace{\left\{ T \geq 0 \mid T\mathbf{1} = p, \ T^\top \mathbf{1} = q \right\}}_{\text{feasible set}}. \tag{2}
$$

enforces that the total "mass" leaving each point $i$ in $X$ equals $p_i$ and the total arriving at each point $j$ in $Y$ equals $q_i$. Inside the double sum, the term $|C_{1,ik} - C_{2,jl}|^2$ quantifies the cost when the distance between $i$ and $k$ in $X$ is paired with that between $j$ and $l$ in $Y$. The product $T_{i,j}T_{k,l}$ carries exactly how strongly those two pairs of points are matched under the coupling $T$. By optimizing over all such couplings, GW finds the correspondence that best aligns the relational structures of the two spaces without ever requiring them to live in the same metric space.

To compare each participant to the layer-wise representations using GW, we first computed the representational dissimilarity matrices (RDMs) of each participant and each layer. RDMs are matrices $X \in \mathbb{R}^{n \times n}$ representing the correlational cost between each pair of stimuli $(i, j)$. To compute the RDMs, we followed [32] and [15]: we applied PCA to reduce the dimensionality of each raw sequence (fMRI and model activations), standardized the signals ($z$-scoring), computed the squared costs, and then normalized the results by the mean to preserve comparability.

In addition to GW, we also computed the centered kernel alignment (CKA) of each participant to each layer, as well as the overall mean for the [+POS] and [-POS] conditions. CKA allowed to measure how similarly two sets of representations encoded relationships across points with minimal intermediate steps. CKA has been previously used in the context of cognitive science [12], deep learning [29] and alignment of artificial and biological neural networks [33].

We followed the unbiased kernel alignment formulation [29] (Equation 3). The selection of this variant was motivated by previous findings on the limitations of the original CKA implementation (e.g., [33]). The selected variant has proven robust to high-sample low-dimensionality and low-sample high-dimensionality data setups [33].

$$
\text{CKA}(K, L) = \frac{\text{HSIC}(K, L)}{\sqrt{\text{HSIC}(K, L) \, \text{HSIC}(L, L)}}, \tag{3}
$$

### 3.5 Language Specificity

In addition to the explicit brain-LM comparisons, we also conducted additional experiments to ensure the brain scores were *human language specific*. We ran the same brain score pipeline on three additional transformer models. The training objective of these model was the same as that of the linguistic bidirectional models analyzed (fill in the masked token). The difference came from the training data: these control models were trained on protein folding [51]. Given the architectural and training similarities (see Appendix A for comparisons), but obvious data differences, effects on correlational scores would unveil the relevance of linguistic exposure on brain scores. Testing on these models allowed to compare and contrast the meaningfulness of the results obtained from models trained on natural language only.

## 4 Results

### 4.1 Positional Information is Key for Brain Alignment

Figure 1 shows the mean brain-model correlations for each model class under four conditions: PCA-compressed versus original feature spaces, crossed with positional ([+POS]) and non-positional ([−POS]) internal representations.

**Effect of PCA compression.** In the bidirectional models (Figure 1, left), PCA increased brain alignment for [−POS] inputs (paired Wilcoxon, $p < .01$)[4] but produced no reliable change for [+POS] inputs ($p = .54$). Causal models followed the same pattern: a significant boost under [−POS] ($p < .05$) and similarity under [+POS].

**Positional versus non-positional encodings.** Collapsing across dimensionality, every model class showed a highly significant gap between [−POS] and [+POS] conditions ($p < .01$). Bidirectional encoders partially benefited from explicit positional embeddings. However, maximum [−POS] scores, especially with PCA, approached [+POS] performance, consistent with their masked-language objective, which may infer position implicitly [54]. Decoder-only models, lacking future context, remained strongly dependent on explicit positional cues, which resulted into a wider performance gap.

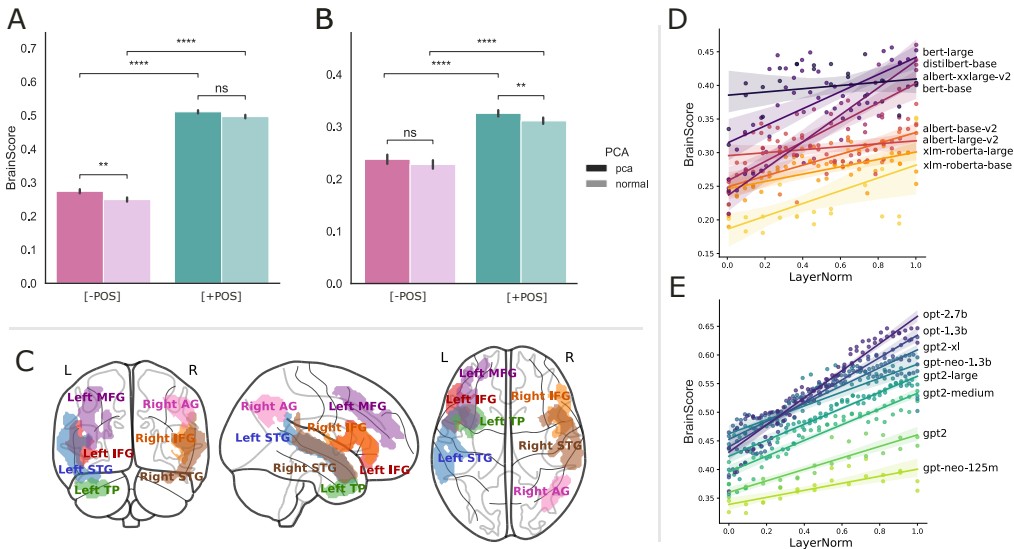

Figure 1: **A-B**: Differences in brain correlations across positional embedding and dimensionality conditions (PCA vs full dimensions) for causal (A) and bidirectional (B) models. **C**: Selected language processing regions of interest (ROI) of each subject. **D-E**: Brain-model correlation trends across model depth for bidirectional (D) and causal (E) models.

**Layer trajectories in bidirectional models.** Patterns diverged across architectures: some displayed a smooth monotonic rising (e.g., `distilbert-uncased`), whereas others dipped in the middle layers before recovering at the deepest (`albert-large-v2`) (see Figure 2). Several models maintained above-average [−POS] scores, pointing to implicit positional inference. All bidirectional encoders peaked in the final layers.

**Layer trajectories in causal models.** Decoder-only models were strikingly consistent: brain scores climbed steadily with depth. Most peaked at the final layers, although others showed global mean maxima mid-stack. All [−POS] curves laid below their corresponding [+POS] counterparts and overall mean.

**Peak correlations.** The highest bidirectional score was $r=.46$ (`bert-large-uncased` and `distilbert-base-uncased`); the highest causal score was $r=.65$ (`opt-2.7b`). These results approximate those reported in most recent work (e.g., [2]).

---

[4]Multiple comparisons corrected with Bonferroni correction. Same criterion applies to all experiments.

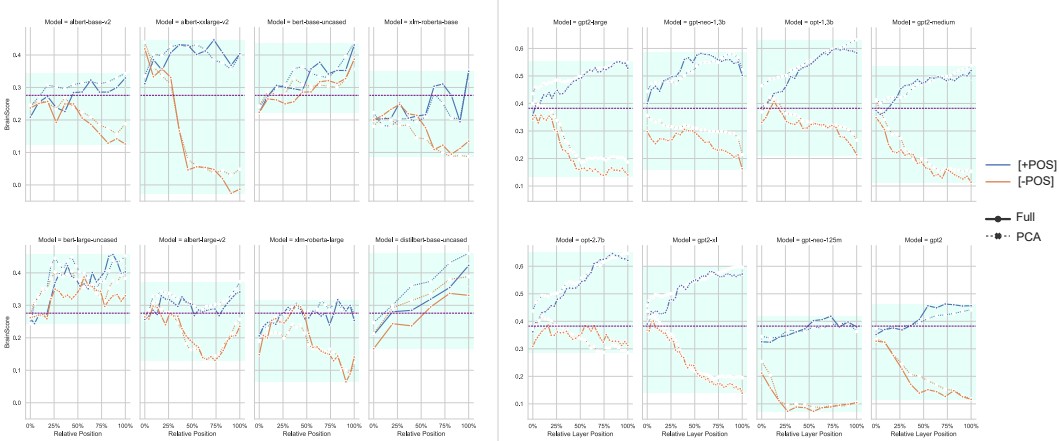

Figure 2: All models' layer-wise brain scores (left bidirectional models, right causal models). The results of the [+POS] condition are shown in blue; orange shows the results for [-POS] condition. Dashed lines show results using PCA data. Purple line indicates overall mean. 0% indicates input layer; 100% corresponds to output layer. Shading indicates range.

## 4.2 Scaled Dot-product Attention Accounts for Some Brain Alignment

To understand how models' internal mechanisms behaved in the observed brain alignment and experimental conditions, we analyzed the scaled dot-product attention heads' brain scores. Analyzing individualized contributions helped clarify whether the observed brain alignment showed distributed properties or specialization clusters within the attention mechanism.

Multi-head scaled-dot product attention is implemented as parallel weight-matrix operations. Let $W_h^Q$, $W_h^K$, $W_h^V \in \mathbb{R}^{d_{model} \times d_k}$ be the query, key, and value projection matrices for head $h \in \{1, \dots, H\}$.

Given a sequence $X \in \mathbb{R}^{n \times d_{model}}$ we compute

$$Q_h = XW_h^Q; \quad K_h = XW_h^K; \quad V_h = XW_h^V.$$

If $\sigma(\cdot)$ denotes a row-wise softmax operation, the attention weights are $\sigma(Q_h K_h^\top / \sqrt{d_k})$. Thus, the result for head $h$ is

$$\text{head}_h = \sigma(\frac{Q_h K_h^\top}{\sqrt{d_{model}}})V_h \in \mathbb{R}^{n \times d_k}.$$

All $H$ heads are concatenated and projected with $W^0 \in \mathbb{R}^{H d_k \times d_{model}}$, producing the final multi-head attention output [50].

Head-wise brain scores were computed as in the layer-wise analysis (§3.3 and §4.1). Figure 3 reports scores for the [-POS] and [+POS] conditions.

**Bidirectional encoders.** For all bidirectional models, head-wise scores dropped markedly when positional information was removed, mirroring the layer-level pattern. Some architectures showed a smooth depth-wise decay (lighter to darker hues along the $x$-axis), whereas others exhibited fluctuations across heads. The trend reversed (i.e., scores increased) once explicit positional encodings were reinstated.

**Causal decoders.** Causal models displayed low, flat scores under the [-POS] setting. Adding positional encodings yielded a substantial boost, yet the improvements concentrated on subsets of heads rather than uniformly across the model. Causal models seemed to follow an idiosyncratic trend rather than a smooth color gradient.

These divergent behaviors align with the different tasks optimized during training. Bidirectional encoders, trained at masked-language modeling, can infer relative position from bidirectional context,

making explicit encodings beneficial but not indispensable. Decoder-only models, optimized for next-token prediction, lack access to future tokens and must therefore rely on externally supplied positional signals. This asymmetry surfaces in the bright-to-dark fade-out of the bidirectional heatmaps versus the largely uniform dark heatmaps of causal models under the [-POS] condition.

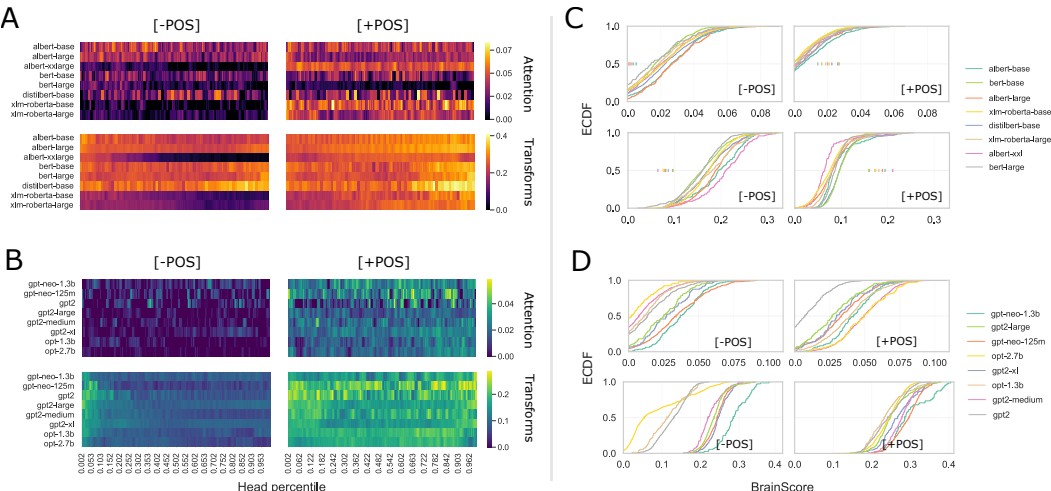

Figure 3: Comparison of model-brain alignment across representational levels and architectural configurations. **A-B**: Average BrainScores for each model under different representational components. The figures show how bidirectional (top) and causal (bottom) architectures differ in their alignment with neural responses. **C-D**: Empirical cumulative distribution functions (ECDFs) of head-level BrainScores, comparing the distribution of brain alignment across individual attention heads for each model (top bidirectional; bottom causal). Curves positioned further to the right indicate that a greater proportion of heads exhibit higher correlations with brain activity. Across panels, causal models consistently outperform bidirectional models, and the inclusion of positional encoding tends to improve model-brain correspondence, suggesting that positional information enhances the representational similarity between transformer models and neural data.

## 4.3 Attention Weights Are Marginally Relevant for Brain Alignment

To account for the impact of raw attention weights on the brain scores we isolated the self-attention operation from the scaled dot-product attention mechanism. To obtain attention weights, the model performs matrix multiplication between the query matrix $Q \in \mathbb{R}^{n \times d_k}$ and the transpose of the key matrix $K^\top \in \mathbb{R}^{d_k \times n}$, which then is scaled by $\sqrt{d_k}$ for numerical stability. The resulting similarity matrix is then passed through a softmax so that its entries lie in $[0, 1]$. As in previous steps, we computed the brain scores following the same pipeline. The brain scores are shown in Figure 3.

**Bidirectional encoders.** Across all bidirectional models, attention weights' brain scores remained uniformly low in the [−POS] and recovered some relevance at the [+POS] condition. However, adding positional encodings produced only marginal gains, far smaller than those observed at layer-level or scaled dot-product attention analyses. Moreover, the changes did not exhibit a coherent depth-wise trend: correlations fluctuated idiosyncratically from head to head, suggesting that no specific subset of bidirectional heads was consistently responsible for brain alignment.

**Causal decoders.** Causal models displayed a moderate increase in brain scores when positional information was supplied. These gains were not evenly distributed: improvements concentrated in contiguous clusters of heads toward the deeper layers of larger models such as `opt-2.7b` and `gpt2-xl`. The partially hub-like patterns underscored the greater dependence of decoder-only architectures on explicit positional vectors for capturing brain-relevant structure.

## 4.4 Brain and Layer Data Spaces Share Characteristics

To analyze the brain-model representational similarities, we computed the layer-wise unbiased CKA[5] and Gromov-Wasserstein distance. The results are provided for each participant and each layer using the most brain aligned causal language model, `opt-2.7b`.

**Layer-wise CKA Analysis.** All participants' brain representations were severely misaligned when the model was deprived of positional information. Interestingly, most subjects showed a similar pattern: CKA steadily decreased to the 0.25-0.37 range (layer 20 to 25) showing a momentous recovery at the 28th layer, and a final decrease from there to the last layer. Adding positional information drastically changed the trends, showing a largely monotonic improvement until reaching the peak similarities ($\approx$0.61-0.87) at the final layers.

**Layer-wise Gromov-Wasserstein Distance.** Gromov-Wasserstein distances offer a different lens to test the similarity of brain and model linguistic representations. Figure 4 shows the GW traces across layers by condition ([-POS] left; [+POS] right) and participant. When deprived of positional information, participant-brain GW metrics showed a spike in the internal layers. When positional information was injected, GW distances showed a steady decay, converging in lower (better) values.

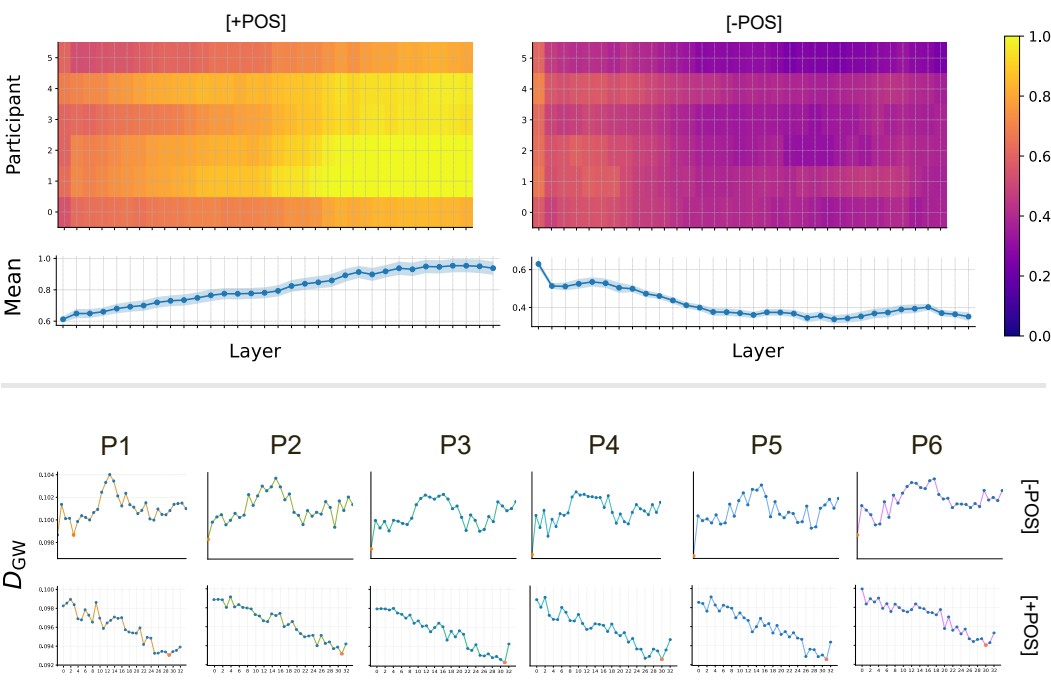

Figure 4: Participant-model CKA (top) and Gromov-Wasserstein distance (bottom) results for [-POS] and [+POS] conditions. For GW distance ↓ is better; for CKA ↑ is better.

## 4.5 Brain Scores are Specific to Language

Figure 5 shows the mean brain scores comparisons for three models trained on non-linguistic tasks and the worst-performing linguistic model (i.e., language model trained on language). All models shared the same masked-token prediction objective; only the training data differed (protein sequences vs text).

**Only Language Training Yields Neural Alignment.** Keeping architecture and learning objective largely constant, substituting natural language with protein folding sequences virtually canceled

---

[5]Unbiased CKA is predictable from brain scores. Still, it provides valuable insights. See Appendix B for post-hoc result analysis.

previously observed model-brain correspondences. The non-linguistic controls obtained brain scores $\leq 0.03$, whereas the poorest text-trained baseline reached 0.23 ($\approx$8-fold higher; $\Delta \approx$0.20; Wilcoxon $p < .01$ for every contrast). These results rule out spurious correlations arising from the task itself and demonstrate that exposure to linguistic input is a critical factor for capturing cortical representations of language processing.

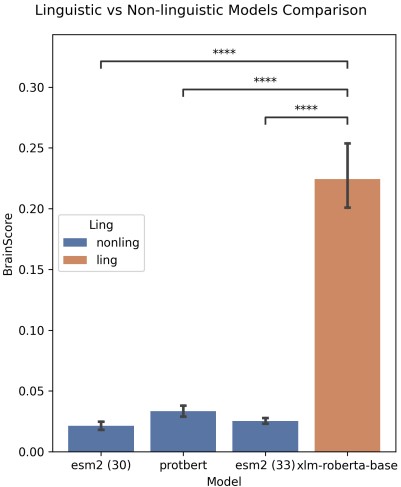

Figure 5: Mean brain scores for non-linguistic control models versus the weakest bidirectional language model, `xlm-roberta`.

## 5    Conclusion

We have confirmed that the internal hidden states of transformer language models are strongly correlated with human brain activity during linguistic tasks. Representations from causal (unidirectional) models predict neural activations significantly better than those from bidirectional models, indicating that their training constraint (word-by-word processing without future context) makes causal models more faithful to human language processing. Correlations with fMRI signals strengthen in deeper layers but vanish when the models, especially those trained causally, are deprived of positional information. These patterns persist after aggressive dimensionality reduction through PCA, ruling out artifacts driven by the curse of dimensionality.

We have also demonstrated that the geometric structure of model and brain representational spaces is jointly organized: removing positional encodings distorts their apparent similarity. CKA confirms the correlation-based findings, while Gromov-Wasserstein distances independently prove (1) the similarity between model and brain linguistic representations and (2) the shared topology of their data spaces. Importantly, both metrics degrade when positional information is ablated.

Finally, we have shown that these effects are language specific. Control models trained on the same objective but non-linguistic corpora achieve significantly lower brain scores under the positional (best performing) condition. Altogether, these results contribute to the broader debate on the plausibility of transformer-based models as biologically plausible models of human language processing.

## Limitations

The findings should be interpreted in light of some constraints. First, the analyzed dataset includes 243 BOLD signals per participant and activations from 19 models. Increasing both the number of subjects and model diversity would increase statistical power and generalizability. Second, the head-wise analysis pooled voxels across cortical regions; no ROI distinctions were drawn. A finer-grained ROI-by-head analysis could reveal differential alignment patterns. Future work should use modality-matched datasets (e.g., spoken-language tasks with audio models, or visual captioning tasks with vision-language models) to disentangle task effects from representational differences.

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

## A  Models

This table provides a detailed description of the model hyperparameters and components. All the disclosed information can be accessed through Hugging Face model configurations.

| Model | Causal | Positional Encoding | Layers | Heads/Layer | $d_{\mathbf{model}}$ | $h_{\mathbf{dims}}$ | Parameters | Authors |
|---|---|---|---|---|---|---|---|---|
| albert-base-v2 | ✗ | absolute (learnable) | 12 | 12 | 768 | 3072 | 11 M | [31] |
| albert-large-v2 | ✗ | absolute (learnable) | 24 | 16 | 1024 | 4096 | 17 M | [31] |
| albert-xxlarge-v2 | ✗ | absolute (learnable) | 12 | 64 | 4096 | 16384 | 235 M | [31] |
| bert-base-uncased | ✗ | absolute (learnable) | 12 | 12 | 768 | 3072 | 110 M | [14] |
| bert-large-uncased | ✗ | absolute (learnable) | 24 | 16 | 1024 | 4096 | 340 M | [14] |
| distilbert-base-uncased | ✗ | absolute (learnable) | 6 | 12 | 768 | 3072 | 66 M | [40] |
| xlm-roberta-base | ✗ | absolute (learnable) | 12 | 8 | 768 | 3072 | 270 M | [13] |
| xlm-roberta-large | ✗ | absolute (learnable) | 24 | 16 | 1024 | 4096 | 550 M | [13] |
| esm2_t30_150M_UR50D | ✗ | rotary (RoPE) | 30 | 20 | 640 | 2560 | 150 M | [52] |
| esm2_t33_150M_UR50D | ✗ | rotary (RoPE) | 33 | 20 | 1280 | 5120 | 650 M | [52] |
| protbert | ✗ | absolute (learnable) | 30 | 16 | 1024 | 4096 | 420 M | [16] |
| gpt-neo-1.3B | ✓ | absolute (learnable) | 24 | 16 | 2048 | 8192 | 1.3 B | [7] |
| gpt-neo-125M | ✓ | absolute (learnable) | 12 | 12 | 768 | 3072 | 125 M | [7] |
| opt-2.7b | ✓ | absolute (learnable) | 32 | 32 | 2560 | 10240 | 2.7 B | [56] |
| opt-1.3b | ✓ | absolute (learnable) | 24 | 32 | 2048 | 8192 | 1.3 B | [56] |
| gpt2 | ✓ | absolute (learnable) | 12 | 12 | 768 | 3072 | 117 M | [9] |
| gpt2-large | ✓ | absolute (learnable) | 36 | 20 | 1280 | 5120 | 774 M | [9] |
| gpt2-medium | ✓ | absolute (learnable) | 24 | 16 | 1024 | 4096 | 345 M | [9] |
| gpt2-xl | ✓ | absolute (learnable) | 48 | 25 | 1600 | 6400 | 1.5 B | [9] |

Table 1: Architectural details for all model families analyzed. $d_{\text{model}}$ = hidden/embedding size; $h_{\text{dims}}$ = intermediate (FFN) size.

## B  Independence of Empirical Results

The following table includes the results from regressing the results from the three experiments conducted.

| | Coef | Std err | t | p | Significance |
|---|---|---|---|---|---|
| Intercept | 2.05 | 0.10 | 0 | 1 | ns |
| CKA | -0.89 | 0.10 | -8.73 | $p < 0.01$ | *** |
| GW | -0.17 | 0.10 | -1.66 | 0.1 | ns |

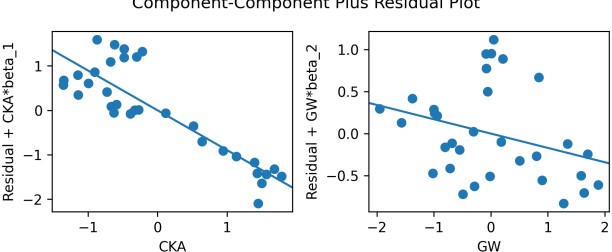

Figure 6: Results for the OLS regression of the experimental results. Adjusted $R^2 = 0.71$.

## C  Data and Code Availability

The code and data used during the experiments are made available here. The GitHub repository includes instruction on how to proceed in order to reproduce the disclosed results.

Code and datasets available upon acceptance.

