# OpenReview forum: "Neural Correlates of Language Models Are Specific to Human Language"
_NeurIPS.cc/2025/Workshop/UniReps — UniReps2025_

### Official Review · Reviewer_TBjV · 2025-09-09
**Neural Correlates of Language Models Are Specific to Human Language**

**Confidence:** 4

**Review:**

This paper presents a well presented and compelling and methodologically robust study examining the correspondence between the internal representations of language models and human brain activity during linguistic processing. It builds upon prior work by testing whether this alignment is (i) robust to dimensionality reduction, (ii) sensitive to architectural components such as positional encoding, and (iii) specific to models trained on human language.

STRENGHTS:
 - The paper addresses well-defined questions that have been under-explored in prior literature, particularly regarding the specificity of brain-model alignment to linguistic training and the role of architectural features such as positional encodings.

 - The author evaluates 19 transformer models across bidirectional and causal architectures.
Employs both traditional brain-score correlations and geometric/topological metrics (CKA, Gromov-Wasserstein distances).
Includes robust controls (e.g., PCA-based dimensionality reduction and protein-trained models).

- The use of protein-trained models, matched in architecture and training objective but not trained on natural language, provides a clean and elegant control for disentangling task-specific from domain-specific effects.
Positional Encoding Ablation

- The finding that removing positional encodings drastically reduces alignment is novel and significant, especially as it underscores architectural dependence for brain-like representations.

- Limitations are clearly stated

WEAKNESSES
- The study relies on fMRI data from only 6 participants and 243 sentence stimuli per subject. While consistent with prior literature, this limits statistical power and may affect generalizability.
- aggregation of ROIs could obscure regional variation in alignment
- While correlation coefficients, CKA, and GW distances are informative, their practical interpretability (e.g., what does an r = 0.46 really mean for language processing fidelity?) remains somewhat abstract.

SUGGESTIONS FOR IMPROVEMENT:
- Broader Biological Interpretation: A brief discussion on whether specific alignment patterns relate to known cognitive functions (e.g., syntactic prediction, semantic integration) could better ground the findings in neurobiology.
- Providing guidance on how to interpret correlation magnitudes and CKA values (e.g., using comparisons with inter-subject fMRI similarity) would help readers better assess effect sizes.

**Score:**

4

**Topic Fit:**

3

---

### Official Review · Reviewer_d6iW · 2025-09-16
**Language representation in LLM and brain**

**Confidence:** 2

**Review:**

The author validated previous results on language representations in LLM and brain responses (fMRI).
The author added several interesting control experiments which I found quite interesting.
The paper is well-presented. Several minor issues I found:\
L47 Defined ReStA but never used it again.\
L48 Mentioned RSA but never defined it.\
Font size in all Figures is too small to read.

**Score:**

3

**Topic Fit:**

3

---

### Official Review · Reviewer_XFP8 · 2025-09-17
**A nice study but I cannot fully agree with the interpretations**

**Confidence:** 3

**Review:**

This is a well-conducted study with a clear pipeline and results that address an important question. However, the authors’ claim that the findings demonstrate that transformer-based language models encode information in a way comparable to the brain is not fully supported by the evidence presented. In particular, we consistently observe that even when training inputs are altered—for example, when positional information is removed—there remains substantial convergence. While the reported correlations decrease, they do not diminish to the extent that would indicate an absence of information. Similarly, tests using non-linguistic control inputs reveal a comparable pattern, though to a greater degree.

**Score:**

3

**Topic Fit:**

3

---

### Official Review · Reviewer_GLmB · 2025-09-17
**Comprehensive analysis of drivers of similarity between LLM and language representation in the brain**

**Confidence:** 5

**Review:**

This paper presents a comprehensive investigation into the factors contributing to the measured similarity between LLM representations and neural responses to language. This work is well-situated within the current debate in the NeuroAI community, where conflicting claims exist regarding the significance and origins of this brain-model correspondence.

Strengths
-The study's primary strength is its systematic approach. The authors methodically disentangle the contributions of multiple factors:  model architecture, input features (e.g., positional information), layer-wise correspondence, the choice of alignment metric, and importantly a comparison between language and non-language data.

-The results are presented with clarity, and the interpretations are well-supported by the empirical findings. The choice of alignment metrics, which adeptly captures both local and global structure, is particularly effective.

- A notable and appreciated contribution is the inclusion of protein-sequence models as a control. This cleverly demonstrates that high performance on a sequence prediction task, in itself, is insufficient to drive brain-like representations, adding significant weight to the authors' claims.

Weakness,
My only suggestion is to expand the conclusion to more explicitly elaborate on how these results contextualize and potentially resolve some of the ongoing debates in the field. This would further enhance the paper's impact.

Overall, this is a well-designed and comprehensive study. It provides a valuable contribution by contextualizing and clarifying existing research on the correspondence between LLMs and the brain.

**Score:**

4

**Topic Fit:**

3